

# Impact of insular landscape features on the population genetics of a threatened climbing palm, *Korthalsia rogersii* Becc., endemic to the Andaman Islands

Sarath Paremmal[1,2], Modhumita Dasgupta[3], Sreekumar VB[4] and Suma Dev[1]

[1] Forest Genetics and Biotechnology Division, Kerala Forest Research Institute, Thrissur, Kerala, India
[2] Forest Research Institute (Deemed to be University), Dehradun, Uttarakhand, India
[3] Division of Plant Biotechnology and Cytogenetics, Institute of Forest Genetics and Tree Breeding, Coimbatore, Tamil Nadu, India
[4] Forest Ecology and Biodiversity Conservation Division, Kerala Forest Research Institute, Thrissur, Kerala, India

Corresponding author
Suma Dev, sumadev@kfri.res.in

## ABSTRACT

Despite the critical structural and functional roles of palms in tropical forest ecosystems and their importance in the local economy and livelihood, palms face significant threats from habitat loss and economic exploitation. Many endemic palms on tropical islands warrant conservation strategies aimed at augmenting the existing gene pool to support effective management and long-term protection of genetic diversity. This study investigated the genetic diversity and structure of *Korthalsia rogersii*, a threatened climbing palm (rattan) endemic to the Andaman Islands in the Bay of Bengal, across seven known populations (including recently identified ones) using microsatellite markers. The aim was to formulate informed conservation strategies by understanding how the island landscape influences the population genetic divergence of the species. Although heterozygosity and bottleneck analyses did not reveal significant genetic diversity loss, a positive correlation between population size and the number of observed alleles points to a potential ongoing decline. Moderate to high genetic differentiation was observed between populations, with geographical isolation contributing to divergence, particularly in the Interview island population. Notably, the South Andaman population (Chidiya Tapu) harbours the highest number of private alleles, despite exhibiting low overall genetic divergence, indicating it may serve as a reservoir of lost genetic diversity. Further, the Bakultala population shows significant within-population relatedness and reduced allelic diversity, indicative of genetic isolation and demographic decline. These findings provide preliminary insights into the role of the island landscapes in the Andaman archipelago in shaping population genetic divergence among plant taxa. Effective conservation strategies should target gene diversity, genetic structure and hotspots of unique alleles identified in the study, prioritising both population size enhancement and genetic augmentation to ensure the long-term survival of *K. rogersii*.

## INTRODUCTION

Palms (Arecaceae), with over 2,550 species, have diversified pantropically, functioning as an ecologically vital group in tropical forests, supporting local economies and community livelihoods (*Tomlinson, 2006*; *Dransfield et al., 2008*; *Baker & Dransfield, 2016*). Nevertheless, palm diversity is under threat due to habitat loss, landscape fragmentation, limited adaptability to climate change, economic exploitation, and alien species invasion (*Blach-Overgaard et al., 2015*; *Carvalho et al., 2016*; *Butler & Larson, 2020*; *Bellot et al., 2022*). Attributed to their key ecological and economic roles, as well as the growing threats they face, palms have attracted more attention in ecological, evolutionary, and conservation research than many other plant groups. Population genetic and genomic studies in palms have revealed existing or onset of genetic consequences in deprived and fragmented populations, including reduced adaptive potential under climate change scenarios (*Galetti et al., 2013*; *Carvalho et al., 2015*; *Soares et al., 2015*; *Soares et al., 2019*; *Carvalho et al., 2017*; *Chacón-Vargas, García-Merchán & Sanín, 2020*; *Montúfar, Recalde & Couvreur, 2021*). These insights contribute to the development of well-informed management and conservation strategies that emphasize the conservation of genetic diversity (*Gardiner et al., 2017*; *Allendorf et al., 2022a*). However, many threatened palm groups remain understudied, especially the Southeast Asian lineages, and taxa that have radiated on islands (*Bellot et al., 2022*). Insularity and isolation are the key drivers of palm speciation, resulting in the origin of many endemic species restricted to a single island or archipelago (*Dransfield et al., 2008*; *Cássia-Silva et al., 2020*; *Kuhnhäuser et al., 2025*). Compared to the mainland counterparts, island populations face greater environmental and demographic stochasticity, which can lead to accelerated genetic diversity loss (*Frankham, 1997*; *Frankham, Ballou & Briscoe, 2002*; *Leigh et al., 2019*).

Diminishing populations of several endemic insular palms like *Brahea spp*, *Pseudophoenix lediniana*, *Dypsis ambositrae*, *D. decipiens*, *Washingtonia* spp. *Carpoxylon macrospermum*, *Tahina spectabilis*, *Neodypsis decaryi* and *Lemurophoenix halleuxii* have been found to possess extremely narrow gene pools and pronounced genetic structuring, attributed to small population size and restricted gene flow (*Dowe, Benzie & Ballment, 1997*; *Shapcott et al., 2012a*; *Rodríguez-Peña et al., 2014*; *Gardiner et al., 2017*; *Klimova et al., 2017*; *Shapcott et al., 2020*).

Despite the remarkable diversification of palm species in tropical Asia, particularly within the subfamily Calamoideae, conservation studies in the region remain comparatively limited (*Dransfield et al., 2008*; *Bellot et al., 2022*). In this current study, we investigate the population genetics of *Korthalsia rogersii* Becc., a threatened rattan (climbing palm) restricted to the Andaman archipelago in the Bay of Bengal (*Mathew et al., 2007*).

Due to its geographical proximity to southwestern Myanmar, the Andaman archipelago predominantly harbours floral elements characteristic of the Indo-Burma biodiversity hotspot, with limited influence from surrounding biogeographic regions, as exemplified by *K. rogersii* (*Ganeshaiah et al., 2019*). The genus *Korthalsia,* an early-diverging lineage within the tribe Calameae, has undergone major diversification in the Sunda Shelf region and currently comprises 28 species (*Shahimi, 2018*). While the majority of species are

concentrated in this region, a few outliers extend into Myanmar, the Andaman Islands, and further eastward beyond the Wallace Line (*Dransfield, 1981*). The Andaman Islands are home to two species of *Korthalsia*: the widespread *K. laciniosa* across the Sunda Shelf, and *K. rogersii,* a narrow endemic confined to the Andaman Islands and listed as threatened (*Mathew et al., 2007*).

*K. rogersii* was first documented in the collections of C. G. Rogers in 1904 (*Beccari, 1918*). Nearly a century later, the species was relocated in 1993 during the *Flora of India project*, from South Andaman. It is a laterally branching hapaxanthic rattan, characterized by a climbing habitat and clustered stems that flower once and then die (Fig. 1). The species was considered to be limited to three populations: Chidiya Tapu/Burmanallah (South Andaman), Havelock/Shaheed Dweep (Ritchie's Archipelago), and Diglipur (North Andaman). However, during recent field work, four additional populations were identified at Interview Island, Betapur, Bakultala, and Baratang (*Sarath et al., 2025*). Most of the populations comprise fewer than 30 individuals and are confined to evergreen forest patches of the islands in proximity to riparian habitat, tightly co-distributed with *K. laciniosa* (*Sarath et al., 2025*). However, two populations (Interview Island and Chidiyatapu/Burmanallah) were exceptional with a complete absence of *K. laciniosa* and relatively healthier populations of *K. rogersii*, along with evidence of regeneration (*Sarath et al., 2025*). Despite its rarity, *K. rogersii* is a highly valued rattan, harvested primarily for the rattan handicraft industry and recognised as one of the Non-timber Forest products (NTFP) from the Andaman Islands (*Senthilkumar et al., 2014*). Beyond economic exploitation, the persistence of this climbing species depends on intact forest structure. However, its survival has been increasingly threatened by over 150 years of forest degradation due to coupe forestry—a logging-based management practice formerly implemented in the Andaman Islands. Although recent forest policies have shifted significantly towards conservation (*Surendra, Osuri & Ratnam, 2021*), the recovery of previously logged forests to their original ecological integrity remains incomplete. Structural and compositional changes from past logging continue to influence forest dynamics, thereby affecting the long-term viability of the species.

The Andaman Forest Department has recognised *K. rogersii* as threatened and classified it as rare in the regional red list (*Nayar & Sastry, 1990*; *Manohara, Linto & Renuka, 2010*; *India Biodiversity Portal, 2025*). However, comprehensive knowledge of its distribution, ecology, genetic diversity and population genetic characteristics remains limited. Therefore, in this study, we intended to investigate the population genetic characteristics of the species, as such information is crucial for the success of population recovery strategies, including reintroduction, translocation and both *in-situ* and *ex-situ* conservation efforts (*Shapcott et al., 2020*; *Hoban et al., 2021*). We hypothesize that geographical isolation resulting from insularity, such as that of Havelock and the interview islands, along with a patchy or distinct distribution, is likely to have influenced genetic divergence in the species. Furthermore, the fragmented distribution, coupled with population decline, may have resulted in genetic erosion. Insights into genetic diversity parameters, including the identification of genetically

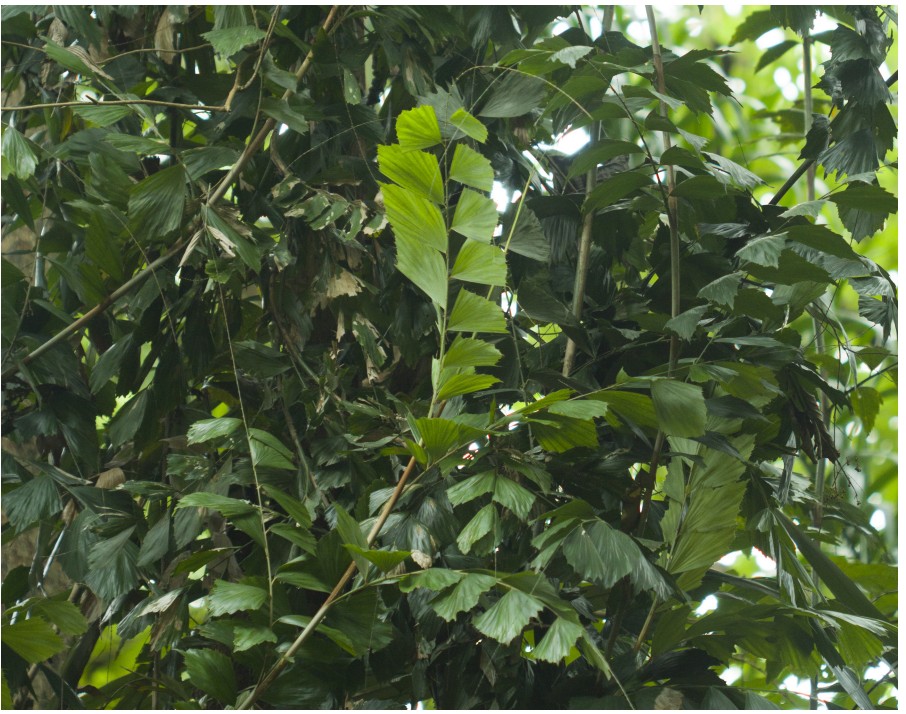

**Figure 1** Habit of *Korthalsia rogersii,* the species is a laterally branching hapaxanthic rattan, characterized by a climbing habitat and clustered stems.

distinct or vulnerable conservation units, will guide the development of targeted strategies for effective resource allocation, management and long-term conservation of *K. rogersii*.

## METHODS

### Sample collection and DNA extraction

All seven known populations of *K. rogersii,* including the newly identified ones, were surveyed during field expeditions in 2022 and 2024 in collaboration with the Andaman and Nicobar Forest Department (Field permit No.F.10 (G-I)/39/Vol.X/652 & No.F.10 (G-I)/39/Vol.XI/462). Leaf samples were collected from the encountered individuals across the populations. To minimize the risk of sampling clonal individuals, especially in small populations (Radhanagar, Betapur, Bakultala and Baratang), sampling from closely stand individuals was avoided. In larger populations (Interview islands, Havelock and ChidiyaTapu), a minimum distance of five m was maintained between sampled individuals. Sample sizes per population ranged from nine to 22, proportional to the population size. Species identification and selection of individuals were guided by detailed morphological descriptions of the species (*Renuka & Vijayakumaran, 1995*; *Mathew et al., 2007*). A total

**Table 1  Population genetic parameters of seven populations of *K. rogersii*.**

| Population | *n* | Ho | He | Ar | Np | Fis | Within population relatedness |
|---|---|---|---|---|---|---|---|
| Interview Island | 18 | 0.64 | 0.66 | 4.476 | 3 | 0.08 | 0.078 |
| Radhanagar | 8 | 0.59 | 0.62 | 4.265 | 2 | 0.078 | 0.073 |
| Betapur | 9 | 0.61 | 0.64 | 4.216 | 2 | 0.129 | 0.068 |
| Bakultala | 13 | 0.64 | 0.59 | 3.842 | 6 | −0.089 | 0.15[*] |
| Baratang | 11 | 0.58 | 0.64 | 4.144 | 3 | 0.109 | 0.026 |
| Havelock | 18 | 0.54 | 0.63 | 4.071 | 3 | 0.149 | 0.059 |
| ChidiyaTapu | 22 | 0.59 | 0.64 | 4.597 | 13 | 0.105 | −0.01 |
| *Overall* | 99 | 0.599 | 0.631 | 4.23 | 32 | 0.08 | 0.06[*] |

**Notes.**

*n*, Number of individuals sampled per population; Ho, Observed heterozygosity; He, Expected heterozygosity; Ar, Allelic richness; Np, Number of private alleles; Fis, fixation index.

Asterisk (*) indicates statistical significance ($p < 0.05$).

of 105 individuals were sampled (Table 1), and collected leaves were silica-dried in the field and subsequently transported to the Kerala Forest Research Institute for genetic analysis.

DNA was extracted using a modified cetyl trimethyl ammonium bromide (CTAB) protocol, which included 2% polyvinylpyrrolidone (PVP) in the extraction buffer, repeated washes with chloroform:isoamyl alcohol (24:1), and extended the incubation with isopropanol up to 30 min to enhance DNA yield and purity (*Doyle & Doyle, 1987*).

## Population genetic analysis of *K. rogersii*

Microsatellite loci, originally developed and standardised for *K. laciniosa* (*Dasgupta et al., 2021*), were screened for cross-amplification in *K. rogersii*. Of these, 16 loci were cross-amplified; however, only seven loci exhibited clear, unambiguous amplification and polymorphism during an initial screen of 14 individuals (Table S1). These seven loci were selected for SSR genotyping across 105 individuals. Forward primers were labelled with 6-Carboxyfluorescein (6-FAM)/ Hexachlorofluorescein (HEX) and used for PCR amplification. Amplicons were analysed using an ABI 3730XL sequencer (Applied Biosystems) with Gene Scan 500LIZ (Applied Biosystems) as the internal size standard. Detailed PCR conditions, including reaction components and annealing temperatures, for each locus are provided in Table S1. Allele sizing and scoring were performed using GENEMAPPER software v4.0 (Applied Biosystems).

## Genetic diversity analysis

The genotype data set was filtered for loci and individuals with >20% missing data. Duplicate genotypes were removed, and unique genotypes were retained for further analysis. MICRO-CHECKER was used to estimate the presence of null alleles and large allele dropout (*Van Oosterhout et al., 2004*). Chi-squared and exact tests were used to test Hardy–Weinberg equilibrium (HWE), as implemented in the *Pegas* package in R (*Paradis, 2010*). Summary statistics, including the number of alleles per locus per population, allelic richness (Ar), number of private alleles (Np), observed heterozygosity (Ho), expected

heterozygosity (He) and fixation index (Fis) were estimated. Data filtering and summary statistics were carried out using the *Adegenet* package in R (*Jombart, 2008*).

Since the Fis is also influenced by null alleles or other deviations from HW, we also explored different inbreeding coefficients, (*Li & Horvitz, 1953*), as modified by *Ritland (1996)* (LH), and *Lynch & Ritland (1999)* (LR) using the *related* R package (*Pew et al., 2015*). We used the *grouprel()* function from the related R package to estimate the average pairwise relatedness within each population. To evaluate whether the observed within-group relatedness was greater than expected by chance, the function performed random permutations of individual group assignments, maintaining the original group sizes. Within-group relatedness was recalculated for each permutation, repeated 10,000 times. This generated a null distribution of expected relatedness values under random group membership. The observed within-group relatedness values are summarized in Table 1.

## Population genetic structure

Pairwise population differentiation (Fst; *Weir & Cockerham, 1984*) was estimated using the *diffCalc* function in the *diveRsity* R package with 1,000 bootstrap replicates (*Keenan et al., 2013*). Individuals were resampled with replacement within populations to generate confidence intervals, and differentiation was considered significant when the lower bound of the 95% interval exceeded zero.

Population genetic structure was further analysed using discriminant analysis of principal components (DAPC) implemented in the *Adegenet* package in R (*Jombart, 2008*). To determine the optimal number of principal components (PCs) to retain, cross-validation was carried out in DAPC using *xvalDapc* function. During cross-validation, the dataset was partitioned into a training set (90%) and a validation set (10%) using stratified random sampling to ensure representation from all populations in both datasets. The DAPC analysis was performed on the training set using varying numbers of retained PCs, and the optimum number of PCs was determined based on the assignment success of individuals in the validation set. The number of PCs that resulted in the lowest mean squared error (MSE) was considered optimal. This optimal configuration was then selected to generate a scatterplot based on the first and second linear discriminants of DAPC, providing a visual representation of genetic clustering among populations.

Bayesian analysis of population genetic structure and admixture patterns was performed using STRUCTURE with a burn-in period of $10^5$ iterations followed by $10^6$ Markov Chain Monte Carlo (MCMC) replications (*Pritchard, Stephens & Donnelly, 2000*; *Hubisz et al., 2009*). For each K value (number of genetic clusters), ranging from 1 to 10, 25 replications were performed to ensure consistency of results. STRUCTURE analyses were carried out with and without the LOCPRIOR model. In the LOCPRIOR model, sampling locations were included as prior information to assist clustering. This approach is particularly useful in the case of data sets that contain relatively few markers, small sample sizes, or very weak signals of population structure that may not be captured by the standard STRUCTURE model (*Hubisz et al., 2009*). To determine the optimal number of genetic clusters (K), we used Structure Selector, which integrates multiple estimation methods (*Li & Liu, 2018*). Given the tendency of the widely used $\Delta K$ statistic to favour $K = 2$, we also

considered alternative metrics such as Ln Pr(X|K), MedMedK, MedMeanK, MaxMedK, and MaxMeanK, as implemented in Structure Selector (*Janes et al., 2017*).

## Isolation by distance (IBD) and environment (IBE)

To better understand the factors shaping genetic differentiation in *Korthalsia rogersii*, we evaluated the influence of geographic and environmental distances among populations. Among the Great Andaman populations (Radhanagar, Betapur, Bakultala, Baratang, and Chidiya Tapu), both geographic distance and environmental variation, particularly latitudinal climatic gradients, are likely contributing to the observed patterns of genetic structure.

To assess the influence of geographical distance and environmental variation on genetic differentiation among individuals in spatially separated populations, Mantel and partial Mantel tests were performed. Pairwise genetic distances among individuals were calculated using *dist.gene* function in the *ape* R package, while a pairwise geographical distances matrix was constructed using *geodist*. The environmental layer of nineteen bioclimatic variables was sourced from CHELSA Version 2.1 (*Karger et al., 2017*). Environmental values at each sampling point were extracted using *raster* package in R, and Euclidean distances between sites were calculated using *dist ()* function. The Mantel test was used to assess isolation by distance (IBD) and isolation by environment (IBE) using geographical distance and environmental distance as predictors, respectively, for IBD and IBE, and genetic diversity as the response variable. For the partial Mantel test, one predictor variable (either geographic or environmental distance) was kept as a covariate to evaluate the independent effect of the other on genetic distance. Both Mantel and partial Mantel tests were carried out using the *vegan* R package, with statistical significance assessed through 10,000 permutations (*Oksanen et al., 2015*).

## Bottleneck test

Recent bottleneck events were evaluated using BOTTLENECK v 1.2.02 (*Piry, Luikart & Cornuet, 1999*). An increase in the heterozygosity (He) over the heterozygosity expected (Heq) at the mutation drift equilibrium was considered an indication of a recent bottleneck event. Given that fewer than 20 loci were used in the analysis, Wilcoxon's signed-rank test was applied, which is recommended for studies with a limited number of loci. The test was run with $10^4$ iterations under both the stepwise mutation model (SMM) and the two-phase model (TPM; and 95% SMM) for the analysis.

## Ecological niche modelling

GPS coordinates and altitudes were documented for each sampled individual during field surveys using a handheld GPS device (eTrex 30; Garmin GmbH, Garching, Germany). In total, 69 occurrence points were recorded. Additional occurrence points were obtained from herbarium collections at the Kerala Forest Research Institute (international acronym 'KFRI'). Nineteen bio-climatic variables for the period 1981–2010 were retrieved from the CHELSA Version 2.1 with spatial resolutions of 30 arcsec (*Karger et al., 2017*). Bioclimate layers were trimmed to the distribution area of each species using the *drawExtent()* and *corp ()* functions in the *raster* R package. To reduce spatial autocorrelation, occurrence

points sharing the same grid (one km$^2$) were thinned using R packages *spThin*, and 15 spatially distinct points retained were used for modelling (*Aiello-Lammens et al., 2015*). Multicollinearity among the 19 environmental layers was assessed using the Variance Inflation Factor (VIF), and highly correlated layers were excluded from further analysis. Remaining independent layers were retained, and Ecological Niche modelling was carried out using the SDM R package (*Naimi & Araújo, 2016*). Initially, we employed the presence-background algorithm (Maxent) to model the potential distribution of *K. rogersii*. However, the resulting model exhibited poor prediction performance, with a low area under the curve (AUC = 0.5). Consequently, alternative presence-absence algorithms, including generalised linear models (GLM), boosted regression trees (BRT), and random forests (RF), were applied using imputed pseudo-absence points. Despite this, none of the models achieved satisfactory predictive accuracy (AUC < 0.8). Among them, the Maxent model performed relatively better, although still below the threshold for reliable interpretation. Therefore, ENM outputs were not used for further inference but are included as a (Fig. S1) for reference.

The lower prediction accuracy is likely due to a combination of factors: the species' highly restricted distribution, small number of spatially independent occurrence points ($n = 15$), and potential microhabitat specificity that is not captured by coarse-scale bioclimatic variables. Rare or patchily distributed species often produce limited occurrence data, which reduces model reliability and increases uncertainty in predictions (*Moudrý et al., 2024*).

# RESULTS

## Population genetics

Microchecker analysis identified a heterozygote deficit in two microsatellite loci, *Kle23* and *Kle18*, indicating null alleles. However, since the null alleles were not consistently detected across all populations, these loci were retained for further analyses. The observed heterozygote deficit appeared to be driven more by population genetic phenomena than by the presence of null alleles. Hardy-Weinberg Equilibrium (HWE) tests, using both Chi-squared and Monte Carlo permutation methods, revealed that five of the seven loci deviated from HWE, indicating population subdivision and/or restricted gene flow (Table S2). However, in both Chi-squared and Monte Carlo permutation tests, no loci consistently deviated from HWE across all populations, nor did any population consistently deviate across all loci (TableS S2A & S2B). Genetic diversity parameters, including the number of private alleles (Np), allelic richness (Ar), observed heterozygosity (Ho), expected heterozygosity (He) and inbreeding coefficient (Fis), are provided in Table 1. Expected heterozygosity and allelic richness were generally comparable across populations, irrespective of sample sizes, except in Bakultala, which showed notably reduced allelic richness (Table 1). In contrast, the number of observed alleles was positively correlated with sample size, likely reflecting underlying differences in population sizes (Fig. S2). The highest number of private alleles (alleles secluded to a single population) was observed in ChidiyaTapu (13), followed by Bakultala (six). None of the populations exhibited significant Fis values, indicating the absence of high inbreeding or deviation from HWE.

Table 2  Fst (*Weir & Cockerham, 1984*) among seven populations of *K. rogersii.*

| Population | Interview Island | Radhanagar | Betapur | Bakultala | Baratang | Havelock | Chidiya Tapu |
|---|---|---|---|---|---|---|---|
| Interview Island | | 58.77 | 37.88 | 42.42 | 88.42 | 100.21 | 153.35 |
| Radhanagar | 0.127* | | 84.62 | 95.59 | 142.86 | 152.1 | 208.31 |
| Betapur | 0.058* | 0.058 | | 14.97 | 59.19 | 67.36 | 124.16 |
| Bakultala | 0.085* | 0.127* | 0.040 | | 47.32 | 59.64 | 112.82 |
| Baratang | 0.085* | 0.039* | 0.012 | 0.049* | | 22.8 | 65.5 |
| Havelock | 0.077* | 0.165* | 0.035 | 0.066* | 0.095 | | 60.58 |
| Chidiya Tapu | 0.106* | 0.085* | 0.005 | 0.035 | 0.014 | 0.070* | |

Notes.
*Lower diagonal: pairwise Fst estimates (*Weir & Cockerham, 1984*); values marked with * indicate significant differentiation based on bootstrap replicates. Upper diagonal: geographic distances (km) between populations.

Estimates of individual inbreeding coefficients (LH and LR) revealed considerable variation, with LH values ranging from −0.43 to 0.79 and LR from −0.38 to 0.63. Within-population relatedness analysis revealed significantly higher kinship among individuals in the Bakultala population compared to expectations under a random mating scenario ($p < 0.0158$), suggesting restricted mating and potential inbreeding within this group. In contrast, most other populations did not show significant deviations from the expected within-group relatedness, indicating more random mating patterns and a lower likelihood of close kin associations. Overall relatedness across all populations was also significantly high ($p < 0.0046$), indicating the presence of population structure. Interestingly, the ChidiyaTapu population exhibited a near-zero within-group relatedness (−0.0069329) and a relatively high number of private alleles.

## Genetic Bottleneck

BOTTLENECK analysis using one-tailed Wilcoxon test ($p < 0.05$) assesses heterozygous excess as an indicator of recent bottleneck events. In this study, none of the populations showed significant evidence of a recent bottleneck event (Table S3). However, the ChidiyaTapu population showed significant deviations from mutation-drift equilibrium in the two-tailed Wilcoxon test, indicating the possible influence of other demographic events.

## Population genetic structuring

Pairwise Fst values ranged from 0.005 (between Betapur and ChidiyaTapu) to 0.165 (between Havelock and Radhanagar) (Table 2). Among the studied populations, Radhanagar, Havelock and Interview Island consistently exhibited moderate to high Fst values in comparisons with other populations, indicating a relatively high level of genetic differentiation and suggesting their genetic distinctiveness. In contrast, Betapur exhibited weak or non-significant differentiation with most populations, except Interview Island. Likewise, Chidiya Tapu showed largely non-significant differentiation with central and southern populations, except with Havelock, where significant differentiation was observed.

Discriminant Analysis of Principal Components (DAPC) revealed clear genetic distinctiveness of the interview Island and Radhanagr populations. The Havelock

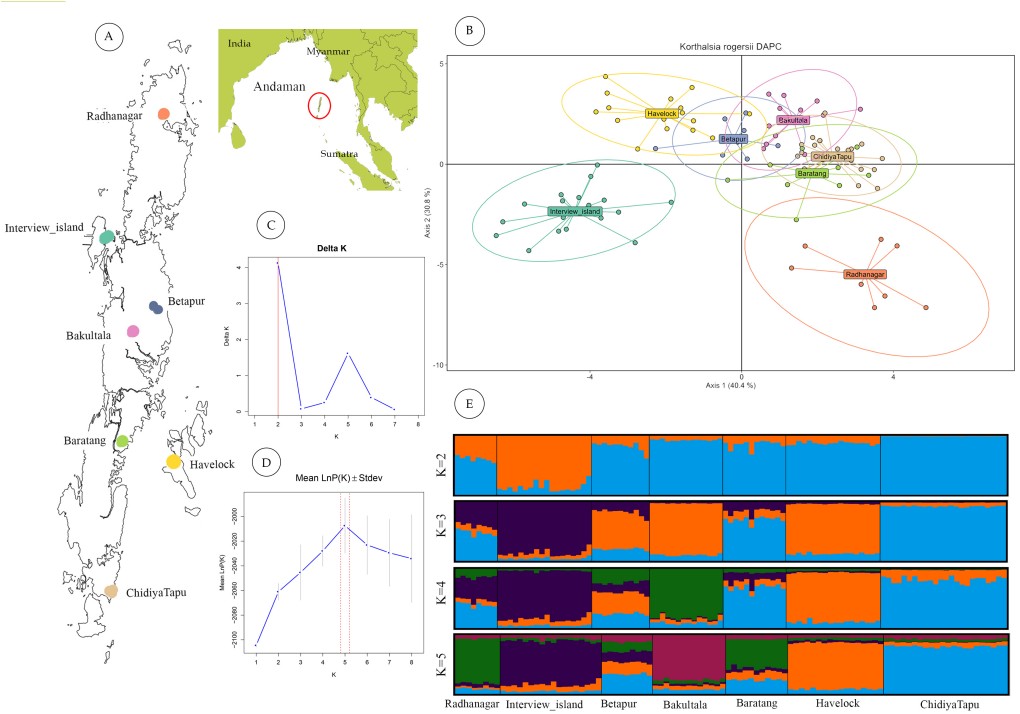

**Figure 2  Geographic sampling and genetic structure of *Korthalsia rogersii*.** (A) Geographic location and landmass of the Andaman Islands showing sampled populations. (B) Genetic structuring revealed by DAPC, highlighting the genetic distinctiveness of the Radhanagar and Interview populations. (C) and (D) Estimation of the optimal number of genetic clusters using the Δ K method and the mean log probability of the data [Ln Pr(X|K)], respectively. Additional estimators (MedMedK, MedMeanK, MaxMedK, and MaxMeanK) also supported $K = 5$; plots for these methods are shown in Fig. S3. (E) Individual admixture proportions based on STRUCTURE analysis for $K = 2$ to 5.

population, though overlapping with the Betapur population, also showed genetic distinctness to a certain extent. In contrast, populations from the Great Andaman Island (Betapur, Bakultala, Baratang and Chidiyatapu) clustered closely together, except for Radhanagar, which remained genetically distinct (Fig. 2).

STRUCTURE analysis without LOCPRIOR model failed to detect distinct genetic clusters, suggesting weak overall genetic structure and a largely panmictic population across the archipelago. This lack of distinct clustering identified may be partly attributed to the limited number of microsatellite loci employed (*Hubisz et al., 2009*). However, when the sampling location was incorporated as prior information using the LOCPRIOR model, a more defined genetic structure emerged. Based on the ΔK method (*Evanno, Regnaut & Goudet, 2005*), $K = 2$ was identified as the most likely number of clusters (Fig. 2). This analysis revealed distinct admixture patterns, particularly in the Interview Island and Radhanagar populations, consistent with the findings from the DAPC analysis.

At higher K values ($K = 5$), supported by the mean log-likelihood (LnP(K)) and additional statistical estimators (MedMed, MaxMed, MaxMean), structure analysis revealed finer-scale population structure consistent with the geographic separation among

populations. At $K = 5$, Radhanagar, Interview Island, Havelock, Bakultala, and Chidiyatapu populations exhibited distinct genetic structure with unique admixture patterns (Fig. 2). In contrast, the Baratang and Beetapur populations showed varying degrees of admixture from multiple clusters, suggesting shared ancestry with genetically distinct groups. This pattern of mixed ancestry is likely influenced by sample size or increased model complexity at higher K.

Among all populations, Interview Island emerged as the most genetically distinct population, showing separation as early as $K = 2$. Havelock Island and Bakultala populations also demonstrated pronounced genetic structuring in the admixture plot at $K = 4$, indicating their genetic discreteness. While most populations from Great Andaman Island (Radhanagar, Betapur, Baratang & Chidiya Tapu), except the Bakulta, appeared admixed at lower K values, a clear, distinct cluster was evident at $K = 5$, particularly in the Radhanagar and Chidiya Tapu populations.

### Isolation by distance and environment

Along with the strong influence of restricted gene flow due to the water barriers, both Mantel and partial Mantel tests indicated that environmental factors also play a role in shaping population genetic differentiation. Among the predictor variables, environmental variation showed a weak but significant correlation with the genetic distance ($r = 0.221$, $P < 0.001$), while geographical distance exhibited no significant correlation ($r = 0.0376$, $P = 0.1$). These results suggest that environmental (climate) factors, although modest, may play a role in shaping the genetic structure of *K. rogersii*.

## DISCUSSION

This study investigated the population genetic diversity, potential bottlenecks, and genetic structure of *K. rogersii*, a threatened and economically exploited rattan species endemic to the Andaman Islands, across its seven known populations. The primary objectives were to delineate genetically distinct and potentially vulnerable populations and to elucidate the influence of the insular landscape of the archipelago and environmental factors in shaping genetic diversity patterns. Despite variations in population size, all surveyed populations exhibited comparable levels of genetic diversity and a differential distribution of private alleles. Furthermore, most populations displayed unique but shallow admixture patterns, likely shaped by a combination of geographic isolation (water barriers), environmental heterogeneity, and within-population relatedness.

### Genetic diversity and population bottlenecks

Populations of *Korthalsia rogersii* exhibited moderate levels of genetic diversity, with expected heterozygosity (He) ranging from 0.59 to 0.66. Despite its restricted distribution and small population sizes, the species has retained considerable genetic variation. While population size is typically a major determinant of genetic diversity, He in this study showed no strong correlation with population size, a pattern also observed in other long-lived plant species (*Kramer et al., 2008*). Moreover, He alone is not considered a reliable predictor of population viability or extinction risk (*Schmidt et al., 2023*). Similar

trends have been reported in other threatened palms, where substantial genetic diversity persists despite conservation concerns (*Shapcott et al., 2007*; *Asmussen-Lange, Maunder & Fay, 2011*; *Shapcott et al., 2012b*).

In contrast, the number of observed alleles, a more sensitive indicator of recent demographic changes, was positively correlated with population size (*Allendorf et al., 2022b*), suggesting that smaller populations of *K. rogersii* may be undergoing partial genetic erosion. Allelic richness (Ar), which reflects the impacts of population size, fragmentation, and habitat degradation (*Browne, Ottewell & Karubian, 2015*; *Carvalho et al., 2015*; *da et al., 2017*; *Soares et al., 2019*), ranged from 3.842 in Bakultala to 4.597 in Chidiya Tapu. However, Ar did not consistently align with population size. Notably, Bakultala, the population with the lowest Ar, exhibited significantly elevated within-population relatedness, suggesting reduced genetic variability. In contrast, populations such as Chidiya Tapu, which had the highest Ar, showed no evidence of significant relatedness.

Despite this high relatedness, Bakultala had a low fixation index (Fis), potentially reflecting a recent bottleneck in which reduced numbers of mating individuals increased relatedness without causing deviations from the Hardy-Weinberg equilibrium. Individual-level inbreeding coefficients showed wide variation across populations (LH: $-0.43$ to $0.79$; LR: $-0.38$ to $0.63$), with no consistent pattern of inbreeding. This range indicates substantial heterogeneity in mating behaviour. Given that *K. rogersii* is monoecious and the presence of self-incompatibility mechanisms is unknown, occasional selfing is likely and may explain the high inbreeding coefficients ($>0.25$) observed in certain individuals, even though population-level inbreeding was generally weak.

Private alleles can indicate restricted gene flow or local adaptation (*Slatkin, 1985*). Their distribution varied across populations and did not correlate with population size. Chidiya Tapu had the highest number of private alleles, although at low frequencies. Despite low pairwise genetic distances with nearby populations, the presence of numerous private alleles suggests Chidiya Tapu may be a relic population retaining ancestral genetic variation. This may be linked to the relatively stable evergreen habitat in southern Andaman and reflects patterns observed in glacial refugia and ancestral populations of other species (*Comps et al., 2001*; *Márquez-Márquez et al., 2025*). Interestingly, such elevated private allele richness in the southern population was not observed in the Andaman day gecko, a species with different habitat preferences and life history traits (*Mohan et al., 2020*).

Bakultala, despite its small size and increased relatedness, harboured the second-highest number of private alleles, indicating both isolation and unique genetic variation. Surprisingly, the geographically isolated populations of Interview and Havelock Islands did not exhibit particularly high private allele richness. This may be explained by historical land connections during the Last Glacial Maximum (*Smith & Sandwell, 1997*), which likely facilitated past gene flow and reduced long-term isolation.

## Population genetic differentiation, admixture and influence of IBD and IBE

Although there is no universally accepted threshold for interpreting Fst values, values $<0.05$ indicate low, $0.05$–$0.15$ as moderate, $0.15$–$0.25$ as high, and $>0.25$ as very high

genetic differentiation (*Hartl & Clark, 1997*). Based on these, *Korthalsia rogersii* populations exhibited a range from low to high pairwise genetic differentiation. Notably, Interview Island and Radhanagar consistently displayed moderate to high Fst values in comparison with all other populations, suggesting substantial genetic distinctiveness. In contrast, the central population (Betapur) exhibited weak or non-significant differentiation from most other sites, suggesting higher levels of gene flow. However, Betapur showed significant differentiation from Interview Island, despite its geographic proximity, likely due to isolation by water. Similarly, Chidiya Tapu was largely undifferentiated from nearby central and southern populations, except for Havelock, where significant differentiation was observed. These patterns underscore the strong influence of intervening water channels in shaping the population structure of *K. rogersii*, a conclusion further supported by both DAPC and STRUCTURE analyses.

Although Chidiya Tapu harbours the highest number of private alleles, it consistently clustered with Middle and South Andaman populations in both DAPC and lower K values of STRUCTURE. This suggests a largely shared gene pool of the Chidiya Tapu population. The presence of rare alleles in the absence of marked genetic divergence supports the interpretation that Chidiya Tapu may represent a relic or ancestral population, retain unique lost genetic variants, while remaining part of a broader gene pool (*Slatkin, 1985*; *Excoffier & Ray, 2008*).

Most South Andaman populations clustered closely together, although Havelock Island appeared somewhat distinct in both DAPC and STRUCTURE analyses, likely a result of its geographic isolation. Interestingly, the Bakultala population exhibited a unique admixture profile despite its low pairwise Fst with the nearby Betapur population. This distinct pattern may be attributed more to high within-population relatedness than to true genetic divergence. Similarly, Chidiya Tapu only formed a distinct cluster at higher K values in STRUCTURE, reflecting shallow differentiation likely driven by its high frequency of private alleles (*Slatkin, 1985*).

Overall, the genetic structure of *K. rogersii* appears to be shaped primarily by restricted gene flow, particularly the water barriers isolating populations on Interview Island and Havelock. In addition to physical isolation, latitudinal climatic variation also seems to contribute to the observed differentiation, as indicated by Mantel test results. Two major genetic clusters were detected within Great Andaman Island, a northern cluster (*e.g.*, Radhanagar) and a southern one (*e.g.*, Chidiya Tapu), with signs of genetic admixture in the central populations (*e.g.*, Betapur and Baratang). Populations on the Interview and Havelock Islands remained distinct from these mainland clusters, likely due to prolonged geographic isolation.

Climatic gradients may further reinforce this structuring; the northern Andaman experiences more pronounced precipitation seasonality than the south, potentially influencing local adaptation. Comparable patterns of north–south genetic structuring have been documented in other insular species, such as the Andaman day gecko (*Phelsuma andamanensis*), which forms two natural genetic clusters (*Mohan et al., 2020*). By contrast, the Andaman keelback (*Xenochrophis tytleri*) shows stronger longitudinal differentiation, driven mainly by saltwater barriers (*Mohan, Swamy & Shanker, 2018*). These differences

suggest that the direction and strength of genetic structuring depend on species-specific life history traits and dispersal capacities, although a general latitudinal trend is evident across the landscape. Limited knowledge of the pollination and seed dispersal mechanisms in *Korthalsia rogersii*, however, hinders a more comprehensive interpretation of the observed genetic patterns. Although empirical data on seed dispersal are unavailable, birds are likely the primary seed dispersers, as observed in many other rattan species. Similarly, pollen dispersal is presumably insect-mediated (*Renuka, Indira & Muralidharan, 1998*). Consequently, gene flow in this species is likely influenced by the movement and behaviour of these biotic vectors.

To comprehensively understand the processes shaping genetic variation in the Andaman archipelago, studies from multiple taxa are essential, along with insights into species ecology, including pollination and seed dispersal mechanisms. Future research should integrate high-resolution genomic data, extensive spatial sampling, and environmental variables to capture fine-scale patterns of phylogeographic structure and landscape-level genetic connectivity in this unique insular ecosystem.

## IMPLICATIONS FOR THE CONSERVATION OF *KORTHALSIA ROGERSII*

Genetic diversity is fundamental to the survival, adaptability, and long-term persistence of species and is increasingly being integrated into conservation frameworks worldwide (*Hoban et al., 2020*; *Hoban et al., 2025*; *McLaughlin et al., 2025*). Although this study used a relatively small number of loci, the findings provide important insights for the conservation of *Korthalsia rogersii*, a threatened rattan species endemic to the Andaman Islands. With most populations being small and potentially at risk, conservation efforts should prioritize increasing population sizes and preserving genetic variation. The observed loss of allelic diversity in smaller populations highlights the need for carefully designed genetic augmentation strategies. However, such interventions must account for the underlying population structure. The Interview Island population, due to its pronounced genetic distinctiveness, should be treated as a unique evolutionary lineage and conserved independently. Havelock Island also exhibits genetic differentiation, though moderate overlap with Middle and South Andaman populations suggests historical gene flow, making it a candidate for cautious integration in restoration plans. The Bakultala population, showing high within-population relatedness, may be susceptible to inbreeding depression and should be closely monitored. In contrast, the Chidiya Tapu population, while genetically aligned with Middle and South Andaman groups, contains the highest number of private alleles and may serve as an important genetic reservoir for augmentation within these regions. These results underscore the need for a genetically informed, population-specific conservation approach for *K. rogersii*.

While this study provides valuable insights for conservation planning, the use of neutral microsatellite markers limits the ability to detect signals of local adaptation, which are critical for climate change-informed conservation strategies. Future research integrating genome-wide datasets, broader spatial sampling, and ecological observations will enhance

the robustness of conservation planning and support evidence-based interventions, such as genetic augmentation or assisted migration, to improve climate change resilience.

## ACKNOWLEDGEMENTS

This study is part of the doctoral research of Sarath Paremmal. We also thank the Andaman and Nicobar Forest Department for facilitating fieldwork. Special thanks to Azhar Ali Ashraf, Assistant Professor, Department of Forestry, Sir Syed College, Taliparamba, Kerala, for his assistance in the fieldwork.

### Funding

This work was supported by the KFRI Plan grants (RP863-2023), the Department of Biotechnology (DBT), Govt. of India (No. BT/ PR 29212/ FCB/ 125/ 14/ 2018, 21.02.2019). The funders had no role in study design, data collection and analysis, decision to publish, or preparation of the manuscript.

### Grant Disclosures

The following grant information was disclosed by the authors:
The KFRI Plan grants: RP863-2023.
The Department of Biotechnology (DBT), Govt. of India: No. BT/ PR 29212/ FCB/ 125/ 14/ 2018, 21.02.2019.

### Competing Interests

The authors declare there are no competing interests.

### Author Contributions

- Sarath Paremmal conceived and designed the experiments, performed the experiments, analyzed the data, prepared figures and/or tables, authored or reviewed drafts of the article, and approved the final draft.
- Modhumita Dasgupta conceived and designed the experiments, authored or reviewed drafts of the article, and approved the final draft.
- Sreekumar VB conceived and designed the experiments, authored or reviewed drafts of the article, geographical location of the field samples, and approved the final draft.
- Suma Dev conceived and designed the experiments, analyzed the data, authored or reviewed drafts of the article, and approved the final draft.

### Field Study Permissions

The following information was supplied relating to field study approvals (*i.e.*, approving body and any reference numbers):

Andaman & Nicobar Forest Department.

## Data Availability

The data and code are available at GitHub and Zenodo:

- https://github.com/sarathpi/Popgen/tree/main/Krog%20data

- Paremmal, S. (2025). Impact of Insular Landscape Features and Isolation on the Population Genetics of a Threatened Climbing Palm, *Korthalsia rogersii*, endemic to the Andaman Islands [Data set]. Zenodo. https://doi.org/10.5281/zenodo.15421147.

## Supplemental Information

Supplemental information for this article can be found online at http://dx.doi.org/10.7717/peerj.20265#supplemental-information.

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
