# Peer review of "Impact of insular landscape features on the population genetics of a threatened climbing palm, Korthalsia rogersii Becc., endemic to the Andaman Islands"

_PeerJ, doi:10.7717/peerj.20265_

## Round 0.1 · original submission · Major Revisions

· Academic Editor

Major Revisions

We received mixed reviews from three experts. I think that this study is important as it investigates the population genetics of a threatened Island species. However, some important concerns were raised by the reviewers, particularly regarding the limited number of loci used (see comments from Reviewer 2). The authors must address all comments raised by reviewers 1-3, reanalyze, and revise the paper accordingly. The authors should also provide a point-by-point response to the comments.

**Language Note:** The review process has identified that the English language must be improved. PeerJ can provide language editing services - please contact us at [email protected] for pricing (be sure to provide your manuscript number and title). Alternatively, you should make your own arrangements to improve the language quality and provide details in your response letter. – PeerJ Staff

Reviewer 1 ·

Basic reporting

The manuscript will benefit from professional English editing. Raw data was not shared, or it was not indicated.

Experimental design

I consider the experimental design to be rigorous. However, I have suggested additional analyses that may improve the manuscript.

Validity of the findings

The study is important, as almost no information exists for this species. However, the conclusions and findings are poorly linked to the existing data on other species.

Additional comments

The study uses microsatellite markers to describe genetic diversity and population structure in the endemic climbing palm species. The manuscript is clearly written, and the objectives of the study are explicitly stated. Besides several minor changes, I have suggestions for the additional analysis and changes in the Discussions section. Although the authors discussed their findings, they did not include information on other related species. For example, the genetic diversity is not discussed properly, and no comparison was made with other palm species. Similarly, when discussing population differentiation, authors often overlook the possibility of including similar studies of other palm species or plant species from the same geographic location.
Some of the citations found in the text are not cited in the “References” section. Please revise—example Shapcott et al. 2012b found in text but not in the References.

Line 45 remove “plant”
Line 47 remove “increasingly”
Line 74 Please, restructure the phrase, it is unclear in the current version
Line 85 include “there are 33 palm”
Line 89 remove “only”
Line 95 remove “majorly”
Line 97. End phrase after “K. laciniosa and K. rogersii.” Then begin a new phrase, removing “among them”
Line 161 Please clarify what the “incubation step” means
Lines 271-273 Move this phrase to the Population genetic structure Results.
Line 300. Which of the 19 bioclimatic variables was significantly associated with genetic structure? Did you combine all 19 variables into one?
Line 312. Based on the study's objectives, it will be important and interesting to estimate relatedness among individuals. It will allow to investigate dispersal and mating. I suggest estimating relatedness among individuals.
Line 317 Fis is not equivalent to the coefficient of inbreeding. It may also suggest null alleles or other deviations from HW. I would suggest estimating a different inbreeding index. For example, one from related R package. https://rdrr.io/rforge/related/
Lines 330 and 440. Are the seed and pollen dispersal for the species known? They may be important in explaining levels of diversity, inbreeding, and connectivity among populations. Please include the discussion on dispersal.
Line 342. Gene flow or its absence?
Figure 1. Please include labels for each plot and a corresponding description of each plot. In the current version, it is not clear.
Table 1. Please provide the overall information (for all populations combined) for each genetic diversity index.
Table 3. Please explain how and why you performed the partial Mantel test for Genetic vs. Geographic distance.

Reviewer 2 ·

Basic reporting

The manuscript’s relevance and value are there. It is indeed important to characterize the genetic diversity of the insular populations of this economically important rattan. The methods used, although they are not novel or of general interest, they are correct, standard and adequate. The sequencing breadth (number of microsatellites) is very low for today’s standards. The sampling design was appropriate.
I must say, as a reviewer I was significantly disappointed with how careless the authors were about presenting the results and discussing them. Please see my comments below on the figures and figure legends, no manuscript should ever be submitted before major edits have been caried out.

Experimental design

I will leave this for the chief editor to decide. I think maybe 7 SSR loci would pass some years ago, but they seem insufficient now that we have so much more sequencing power and that genotyping has become so much cheaper. It is also not clear why of the 57 only 16 cross-amplified and then only 7 were used. With SSRs, cross-amplification is very likely, but it takes time and patience to figure out the proper amplification protocols, so to me it seems the number could have been increased by a lot with a little bit more effort.
IBE/IBD Line 208 It is unclear to me why the IBD and IBE analyses were carried out, which hypothesis and prediction do they connect to? I am not saying they are wrong. They just seem to be thrown in there simply because they could be done.

Validity of the findings

The findings are valid although limited. Caveats are not discussed and the results are not well presented with attention to detail.

Additional comments

Figure 1
The colors should match – please assign one color per population, both for the structure plot and for the DAPC. The figure legend is incomplete, it should have A, B, C and indicate what each figure is. You also need punctuation on this legend.
Also, a photo of the plant and habitat should be provided as context, this is a conservation manuscript, and we should see who and what you’re referring to.
On figure 1, we need to see the cross entropy plot to see which K is best and why.
Why did you flip the individual structure plots by population? The orientation means a lot because these axes portray very different information.

Table 1
Please use the legend to explain what each of these abbreviations mean (Ar, Np, …), write out Population instead of Pop. In general, the legend seems to have been done to get it out of the way, please complete it. Do not use colon after “Table 1” replace with a period.
You must indicate which FIS are significantly different from one another.

Table 2
FST is a pairwise comparison, so it is “between” not “among”. The legend says seven populations, but the table includes six. Please capitalize “island” because it is part of the name.
Looking at these values, it seems this is all one big population, perhaps maybe only two (Interview Island + the rest).

Table 3
Again, a very sloppy legend “Mantel test Partial Mantel test…” lacks edition and care. You should write out IBE and IBD and explain what Gen,Geo means

References throughout the manuscript
In green, I highlighted where a reference should be provided to back up the claim/ observation.

Grammar
I am not a native English speaker myself, however I did notice issues in places that I have highlighted (in yellow) throughout the pdf of the manuscript where language edits are required.
The introductory paragraphs are too long (in my perspective) consider splitting or shortening them.

Annotated reviews are not available for download in order to protect the identity of reviewers who chose to remain anonymous.

·

Basic reporting

'no comment'

Experimental design

'no comment'

Validity of the findings

'no comment'

Additional comments

The manuscript titled “Impact of insular landscape features on the population genetics of a threatened climbing palm, Korthalsia rogersii Becc., endemic to the Andaman Islands” explores how the spatial configuration and isolation of tropical island habitats influence the genetic diversity and structure of K. rogersii. By sampling seven populations and genotyping individuals using microsatellite markers, the study provides insights into the patterns of genetic differentiation, gene pool fragmentation, and the conservation value of particular island populations.

Overall, the manuscript is well-structured and presents data that can inform genetic management plans for island palm species. However, I encourage the authors to consider the following points to improve the clarity and impact of the work:

1- Introduction: Although the English is generally good, some paragraphs in the Introduction are overly long and dense. Please consider breaking them into shorter paragraphs to improve readability and make the background information more accessible to the reader.

2- Ecological Niche Modelling: Although this analysis is described in the Methods section, no corresponding results are presented in the Results section. Please clarify whether the modelling was conducted and, if so, include a summary of the main findings or explain why the results were omitted.

3- Although no population showed significant heterozygosity excess, Table S3 lacks clarity. Please improve the legend and clearly define the parameters and meaning of each column to help readers interpret the results more easily.

4- All figure and table legends should be revised for greater clarity. Please ensure that all abbreviations and symbols are defined, and that each legend provides enough information for the content to be understood independently of the main text.

5- Consider including a photo or illustrative figure showing the island landscape or the distinct habitats where Korthalsia rogersii populations were sampled. This would help readers unfamiliar with the Andaman Islands better visualize the environmental context and enhance the overall clarity of the study.

6- Consider including Table S1 in the main manuscript, as it contains important information.

---

## Round 0.2 · Minor Revisions

· Academic Editor

Minor Revisions

I am glad that the authors addressed the reviewers' comments in the revised version of teh manuscript. I encourage them to carry out some minor edits suggested by the reviewer.

Reviewer 1 ·

Basic reporting

The introduction was improved. I liked the thorough description of threats and distribution. Moreover, the aims are clear and well justified.
Line 51 remove “often more so”
Line 127 include the total number of sampled populations.
Line 129 remove “ critically”
Line 130 “closely distributed” change to closely stand or close
Line 158 add citation for R
Lines 202 to 204 this part belongs to the result section
Line 242 citation needed for SDM R package
Line 250. It may be interesting to discuss why species distribution modelling did not work. How did you decide which ecological variables to include/exclude?
Line 324, I could not find results for IBD here, Please include.
Figure 2 is not necessary; I suppose you can include the relatedness estimate in Table 1
Table 2 you should include p values for each Fst
Table 3 is not necessary; these results should be included as text.

Experimental design

no comment

Validity of the findings

The authors should discuss in more detail the limitations of the study.,

---

## Round 0.3 · accepted · Accept

· Academic Editor

Accept

I thank the authors for revising the manuscript to address the minor comments. The current version is suitable for publication.